# Phenomenological Fragments of Trinitarian Discourse: Being, Having, Relating

**Robert J. Woźniak**

Faculty of Theology, Pontifical University of John Paull II in Kraków, ul. Kanonicza 9, 31-002 Cracow, Poland; robert.wozniak@upjp2.edu.pl

**Abstract:** Among the important tasks of Trinitarian theology today is the need to rethink its basic conceptual coordinates. This contribution is a proposal for a phenomenological and existential approach to Trinitarian theology. The starting point is the analysis of three essential existential operators, by means of which the depth of the filial experience of Jesus is expressed. These operators are: being, having, and entering into relations. Their analysis in light of the data of the Gospel narrative allows us to create an interesting conceptual framework for a new articulation of the Trinitarian discourse. The article builds on the conviction that the concrete shapes and modalities of Jesus' life are essential moments of the revelation of the Trinity. Before it is simultaneously closed and opened in concrete historical forms of discourse and in concrete metaphysical concepts, it is accomplished in the categorical decisions, actions, and words of Jesus, in which his filial consciousness is revealed. The ambition of the text is to reintroduce metaphysics into theology, however, from a different perspective than was conducted, for example, by classical scholasticism. It is about the existential recovery of metaphysical potential in theology. Revelation takes place in history and in the concrete of life. The metaphysics that theology needs must realize this and, above all, be up to the task of pointing to the living, historical center of Revelation and all theology. The article argues that such an existential deepening of metaphysics for Trinitarian theology can be conducted through collaboration with phenomenology. In such a perspective, the fragments of Jesus' life, especially his way of being, having, and entering into relations, are ways in which the Trinity reveals itself in history. In this way, Trinitarian theology ceases to be a mere intellectual puzzle, becoming an existential paradigm, and the fragments of Revelation reveal an impressive structure in which speculation and life become integral paths toward the Mystery. On the formal side, the text argues for the integration and use of both metaphysics and phenomenology in Trinitarian theology to enhance its existential impact. This in turn implies a rethinking of how metaphysics, phenomenology, and theology itself are usually understood as well.

**Keywords:** Trinity; Trinitarian theology; metaphysics; phenomenology; being; possession; relation

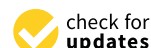



## 1. Introduction

The central truth of faith, which is the inner life of God, is revealed from the perspective of the fragments of the life of Jesus of Nazareth, who is The Christ. This intuition characterizes the whole great tradition of theological thinking, from the writings of the NT through the first great synthesis of Irenaeus, then Thomas and his Christology of the mysteries of Jesus' life, and finally the theology of the history of Urs von Balthasar. The latter even bases his universal vision of the meaning of history on the intuition that its global meaning is revealed and given in small fragments that make up the whole of the eternal life of the Logos in the body: childhood, youth, maturity, death, resurrection, and spousal unity with the Church (von Balthasar 1990, 264nn).

At this point, the reader is due a more detailed explanation of the title of this article and its main purpose. It is, first, an attempt to propose an extension of the set of methods of Trinitarian theology to include the phenomenological method. This is not about replacing

the other methods or suggesting that phenomenology should take the place of metaphysics in theology. The hypothesis of the present text is that phenomenology can be an interesting research option in the space of Trinitarian theology as a complementary way of analyzing the data of Revelation. The present contribution is an attempt to illustrate this hypothesis with the concrete example of selected "fragments" of Jesus' life. But more about that shortly.

For the project proposed here, the fundamental principle of thought is the rule of analogy, as it was formulated at the Fourth Lateran Council in 1215. In this light, any concept of human language applied to speak of God is more dissimilar than similar. Analogy therefore demands respect for the mystery of God. Analogy at the same time opens the field for methodological pluralism in theology, for since no concept exhausts the mystery, we need many methods, many concepts, and many models. And while their multiplication does not diminish God's otherness, it does allow us to better understand the nature and content of His revelation in history. It also helps to speak better, that is, more adequately, about Him. It also contributes to the greater coherence of theology, both in its internal consistency and its openness to interdisciplinarity. The application of elements of the discourse of phenomenology can provide an opportunity to develop new models of Trinitarian life that complement each other.

The subjects of this presentation are three concepts that are the three modal[1] ideas of the Christian vision of the world: being, possession, and relationship. Before they form the fundamental instruments of metaphysical or theological thinking, they already belong to the actual and fundamental structure both of ordinary human life and of the Revelation of the Father, the Son, and the Holy Spirit, deeply inscribed—and this must be strongly emphasized—in the actual apostolic experience, remembered and transmitted in dogma. They are fragments of the Trinitarian discourse and, consequently, the modal concepts, the operators of the Christian vision of reality. These fragments somehow reveal, always keeping the principle of analogy, the inner life of God, which becomes—with the help of these fragments and once again analogically—the unveiling of the truth of the world and its Christian vision.

The choice of these operators is dictated by the fact that they are fundamental dimensions of human life, in which the true nature of existence is revealed. Being, possession, and relation thus have a fundamentally metaphysical value, which can be captured and described using phenomenological tools. At the same time, they remain in close relation to each other: being is a way of possessing oneself, and both being and possessing are determined relationally. One's being and having ultimately reveal a relational nature: they are being oneself and with others and having oneself and others. The conundrum of being and having is only fully revealed in relationships.

The title is a reminiscence of the famous work by R. Barthes, *Fragments d'un discours amoureux* (1977).[2] This conscious borrowing does not mean a simple transfer of the basic ideas of postmodern philosophy to theology. On the contrary, the ambition of the author of this presentation is to liberate it from the pseudometaphysical beliefs of weak thought, which consider the radically fragmented nature of reality and its discourse as its basis and final goal.[3] The borrowing of the title is therefore purely superficial. The understanding of the fragmentation of the Trinitarian discourse is based here more on the intuition of the aforementioned Balthasar, who argued that fragmentation is a carrier of global sense and not a sign of its radical absence. If the metaphor of the fragment is used by postmodern authors to deconstruct the sense, then for Balthasar, the deconstruction of the fragmentary leads to the discovery of the sense. The very fact that God gives Himself to man in the fragments of Jesus' life gives rise to thought. And in a very special way, it makes us think about the meaning of fragmentation as such.

In this respect, the fragments of the Trinitarian discourse echo the classical score of the theo-ontology, whose original shape was determined from the time of the Fathers to the golden age of scholastics in the form of the trinomial of being, truth, and goodness (cf. Bonaventure 1882–1902, pp. 293–16). Of course, they are not to replace it in any way or copy it thoughtlessly, but only to rest on the conviction from which the thinking about

transcendentals emerged. It is a type of thinking that establishes the fragmentation of reality and human discourse from the perspective of a sense of unity. But before we reach this conclusion, we must go through the encounter with the fragments of Revelation, in which God revealed Himself to us and gave Himself to us.[4]

## 2. The Christological-Phenomenological Structure of Trinitarian Discourse: Toward Metaphysics Hidden in Christology

Following J.-L. Marion, one can assume that one of the best methods for studying Revelation is phenomenology[5], which should be understood in the simplest possible way as a return (Husserl 2001, p. 168) to the fact that things give themselves to the extent and in the manner in which they actually give themselves (Heidegger 1996, pp. 25–34). To the extent that phenomenology is based on reduction to donation, it becomes something more than just a superficial science about the appearance of things that underestimates the truth. Phenomenology built on reduction to the things and to their appearance may indeed be a continuation, sometimes a corrective continuation, and never a substitute, of the first philosophy (metaphysics), which studies the most intrinsic nature and structure of things (cf. Rombach 1988, 2010).

According to the logic of the apostolic experience expressed in the Prologue of the first letter of St. John, the revelation of God is phenomenological: it is the sensual, visible, and audible manifestation of the Life in the world (1 John 1:1–2). Christ is the Supreme Revelation of God in the world. Phenomenology applied to Christology means a radical return to *verba et gesta Iesu*. It is from them that the Trinitarian discourse emerges, as they are the mysteries carried out/accomplished in the flesh of Christ (*mysteria perpetrata in carne Christi*). Exploring its nooks and crannies each time means a return to the words, person, and works of Jesus, who is Christ. This fact is confirmed by the judgments of the great christological Councils, which are nothing more than an expression of the words and deeds of Jesus recorded in his experience and apostolic memory.

The foundations of the Trinitarian discourse therefore reside in Jesus[6], as remembered by the apostles.[7] The essential layer of this memory is what they have managed to hear, see, and acknowledge about the identity of Jesus and about how he understood himself. As John the Evangelist puts it frankly, these original facts were not understood by the apostles before Christ's Passover. They became clearer only in the light of the death and resurrection of Jesus.

(1′) Originally, this identity was expressed in how Jesus understood his own existence and being[8]. We should not overlook the fact that in his description of Jesus, John repeatedly puts the expression *ego eimi* on his lips.[9] The power of this expression becomes clear when John's Gospel describes the scene of the capture in the garden. When Jesus identifies Himself in the dialogue with the Jewish guards of the temple by saying "I am", John records that those seeking his capture prostrated themselves in front of him. This culminating gesture is intended to indicate that Jesus expressed in his auto-identification the awareness of the essential connection with the mystery of God's name revealed to Moses on Mount Horeb. In fact, the main reason for Jesus' conviction was the belief of those accusing him that he "made himself equal to God". Jesus is different from all others; after all, "before Abraham became, he already is there" (cf. John 8:58). Jesus' "I am", therefore, possesses a special character—the divine being is revealed in it. We must immediately add that this unveiling is a paradoxical unveiling of the even more paradoxical nature of divine existence, which not only does not abolish the existence of the world and man as in Hegel's *Aufhebung*, but also makes it a sign of itself. Jesus' *am* reveals itself to the maximum in all its greatness in the suffering, stripping, passion, and death: "When you have lifted up the Son of Man, then you will know that I am He" (John 8:28). On the cross, it becomes apparent how Jesus understands his own existence—his original pre-comprehension of being. This is a kind of Paschal metaphysics[10] in which existence is the ability to transcend oneself, to deviate from oneself in a certain way, to go towards the Other. The primary principle of existence is self-giving, the offering of oneself, which does not cause the disappearance

of the subject but constitutes it: the more self-giving, the more existence and being. This Paschal metaphysics marks the first moment of the Trinitarian discourse by reinterpreting the mystery of existence. In this way, in Jesus crucified, the Old Testament name of Jhwh God is confirmed and explained. Fidelity of God ("will be who I am", cf. Exodus 3:14) ultimately reveals itself as the divine being for man (*pro-existentia*), as incomprehensible compassion and care, as being and meaning, and all this is ultimately defined in its meaning by God's fatherhood (Le Guillou 1973). Christ's "I am" is mysteriously inscribed in the existence of the Father: "and the Father are one" (John 10:30). Thus the basic logic of Paschal metaphysics becomes a trinitarian witness: existence is determined by total co-existence, which produces unity. "I am" of Jesus does not enclose him in himself, but co-existence[11] opens him up to the existence of the Father, from whom he comes through eternal birth and through human conception in Mary's womb mediated by the action of the Holy Spirit (annunciation, baptism in Jordan, and Nazareth messianic manifestation). That is why Jesus' understanding of existence presupposes co-existence with the Father and the Spirit. This three-dimensional coexistence (*co-esse*) manifests itself on the cross as the power that "makes everything new" (Rev 21:5).

(2′) The second sign of Jesus' identity is his consciousness of possession.[12] According to the witness of the Gospel and Tradition, Jesus was a poor man: "The Son of man has no place to lay his head" (Matthew 8:20). Most of his disciples come from the poorer strata of society, and those like the tax collector Matthew and Zacchaeus sell a part of their wealth when they decide to follow him. Nevertheless, the original testimony remembers Jesus as the one who has "all that the Father has" (John 16:15). Jesus receives from the Father first all of his life (John 5:26) and certain deeds to do (John 5:36). To have or possess means to share or co-possess. This co-possession mysteriously defines his existence and freedom when he states that "he has his life, he can give it up and he can regain it again" (cf. John 10:18). These passages suggest that Jesus' consciousness of possession stems from his understanding of his own existence in terms of the Paschal metaphysics. In Jesus' own possession of himself, the truth about the pilgrim character of his existence is revealed (see John 1). The cross again becomes the supreme way in which Jesus has everything: in full freedom of self-sacrifice, which at the same time expresses the depth of his existence, he is the holder of the whole. By becoming poor, he enriches us (cf. 2 Cor 8:9). His free possession of Himself comes from sharing everything with the Father. We can say that what Jesus really has is His Father. He is the one who represents the whole of life's wealth for Jesus. He also gives to Jesus every man in possession, so that Jesus may not lose anyone but find everyone (John 17:24 and 18:9). In this sense, one of those found, Peter, states, according to the logic of apostolic witness, that Jesus "has the words of eternal life" (John 6:68), which, as Jesus himself makes it clear, is the result of the revelation from the Father. What constitutes the world of Jesus' possession, as well as the way of possession defined by the radical freedom in which he possesses all that he has (Father, Spirit, life, works, glory, people) and himself is possessed by His Father (filial obedience), becomes in the gospel a fragment of the Trinitarian discourse. In the way Jesus possesses all that he has (*kenosis*), the character of His filial existence is revealed in the world. In this sonship, however, access to the inner life of three persons opens up for the world in order that it may have the fulness of life. This is how Christ's possession is transferred and communicated to the whole world. What Jesus has had from all eternity belongs now to the world.[13]

(3′) The depth of Jesus' being (the meaning of his "I am") and the true nature of possession (the meaning of his *have*) are ultimately visualized and expressed in the affirmation of the relationship as the original and deepest category capable of identifying the identity of Jesus. This category is hidden deep within the structures of the evangelical vision of reality and its Christological infrastructure. Nonetheless, it is radically inscribed in the very center of what the Gospels want to transmit to us. The Paschal metaphysics in terms of existence and possession reaches its peak in setting the center of reality in relationships. Being and possession thus become the *modi* of the relationship, a kind of secondary notion in relation to it. What shines through the life of Jesus and also through his way of possessing is the

relationship. Jesus exists, lives, and possesses in such a way that it becomes visible to him that the relationship is what defines his identity most deeply in all its dimensions. Sometimes this deepest structure of Jesus' existence is overlooked, underestimated, and forgotten. It is difficult not to notice, however, that the basic existential mood of Jesus is to enter consciously and voluntarily into relationships that involve practically everyone, including, in a special way, those on the margins. Jesus' existence is made up of meetings and dialogues in which multiple and varied relationships originate. They flow from the relationship that unites him with the Father and the Holy Spirit and are the way in which the primordial relationship with the Father and the Spirit can extend itself into the world. Jesus came to re-make the world a relational world, so that existence and possession might be determined by relationships with God and others.

From the perspective of the relational universe of Jesus' life, it is worth mentioning Samuel Wells' interpretative proposal (Wells 2015, pp. 11–20, 123–27). He claims that the central theme of the whole gospel and Christian doctrine is what he describes as being with. The life of Jesus is filled with relationships, an outstanding example of which can be found not only in His public activities but also in His hidden life in Nazareth. Without any historical description, without being covered in any transmitted testimony, thirty years in Nazareth are nothing more than a matter of entering into everyday relationships. The longest stage of Jesus' life in the world is filled with simple relations, in which more and more of those two that constitute the whole of his life are revealed: relationships with the Father and the Holy Spirit. In the history of Jesus, everything emerges and returns to them. In the universe of these relationships, the absolute priority is, of course, the relationship of Jesus with the Father: "came from you and came into the world, come out of the world and go back towards you" (cf. J 16:28). It is here, in the relationship to the Father, that a whole world of other relationships begins, none of which compare to this first fundamental one, embedded in the act of giving birth, which constitutes Jesus' esse in its totality.

As one may well have noticed, the evidence in this paragraph comes from the gospel of John. This latest gospel represents, within the New Testament book collection, the most advanced theology and Christology, according to the classic vision of the history of the canon. It is a kind of summit in the development of New Testament theology that collects and articulates its most important intuitions. In this sense, the texts of John's community on which we based our attempt to reconstruct the early Trinitarian discourse are representative of the wealth of the various testimonies of the New Testament authors, who, while presenting Jesus from different positions, come to a great agreement about who he was and what he did. Their witness is consistent with the very similar application of the Paschal metaphysics, in which the being (existence), possession, and relationships of Jesus are a revelation and interpretation of what ultimately means that God is love (1 John 4:16). Thus, the fragments of the Trinitarian discourse are at the same time fragments of the love discourse.

However, what needs to be stressed here in order to summarize this part of our deliberations is the fact that being, possession, and relationship, as described from the point of view of the apostolic witness to Jesus, form an inseparable whole. In the acts of existence, possession, and relationship of which Jesus is the subject, the truth of reality is discovered. The well-ordered structure of existence, possession, and relationship unveiled in the event of Jesus represents the deepest metaphysical truth. In fact, they are fragments of a story about a reality that transcends every known act of existence. As such, they are key and modal concepts of Paschal metaphysics. The true metaphysics of the exodus is not given to us in abstract reflections on the *ipsum esse subsistens* but in the discourse of existence, possession, and relationship so characteristic of Jesus' factual life. It is precisely these concepts that, in the New Testament, become metaphysical categories, operators, or modal concepts of what could be described as its own metaphysics. In the acts of existence, possession, and entering into relationships (being with), of which Jesus is the subject, the divine form of existence is revealed. In them, we are given not only the final Revelation of God in this aeon but also the unveiling of the deepest metaphysical structures of our

own human experience of life in all its fullness. Existence, possession, and relationship are indeed fragments of the Trinitarian discourse in which the divine form of being as a model for the world is given and discovered.[14]

### 3. Between Metaphysics, Phenomenology, and Theology

What was stated until now raises the preliminary but fundamental question of whether metaphysics and phenomenology can be reconciled? An apt answer to the question thus posed was given by L. Tengelyi: "The phenomenological school of thought cannot be reconciled with a metaphysics that seeks to derive fundamental facts of life and the world from first causes and principles. It does not, however, reject any kind of metaphysics for that reason. Our overview has shown that in the last century, different approaches to the phenomenological metaphysics of accidental facticity have emerged. Husserl presented a non-traditional metaphysics of primal facts. In doing so, he envisaged primal facts, which, in contrast to ordinary facts, could be characterized by a factual necessity—the "necessity of a fact". In Husserl's interpretation of primal facts as "primal necessities," the possibility of a metaphysics emerged that was armed against Kant's critique of speculative metaphysics. Although this approach was productively continued almost exclusively by Sartre, related efforts have been reported in the phenomenological tradition. Heidegger's metontological foundation of metaphysics showed parallels to Husserl's metaphysics of primal facts. Over and beyond, she was characterized by her orientation towards a new concept of the world. In French phenomenology, Marion attempted to build another First Philosophy on the "fait accompli" of the appearance of whatever appears. Thinkers like Levinas or Ricoeur, on the other hand, went in a different direction by trying—each in a different way—to clarify the infinite using a phenomenological approach" (Tengelyi 2015, p. 297).

Although in an altered context and object of thought, one can find parallels in this well-documented way of thinking of Tengelyi to the proposals of J. Ratzinger. "For if it is true—contents Bavarian theologian—that the *prae* of God's action is significant for theology, that faith in an *actio Dei* is antecedent to all other declarations of faith, then the primacy of history over metaphysics, over all theologies of being and existence, becomes immediately obvious. It thus becomes obvious that the concept of God is removed from the realm of a mere ousia. It was here that the definitive boundary between the biblical and Greek concepts of God became obfuscated, that this obfuscation was the crux of the repeated patristic attempts to combine Greek thought with biblical faith, and that from this arose for Christian theology a task that is still far from being accomplished. Decisive for the Greek concept of God was the belief in God as a pure and changeless being of whom, consequently, no action could be predicated; his utter changelessness meant that he was completely self-contained and referred wholly to himself without any relationship to what was changeable. For the biblical God, on the other hand, it is precisely relationship and action that are the essential marks; creation and revelation are the two basic statements about him, and when revelation is fulfilled in the Resurrection, it is thus confirmed once again that he is not just one who is timeless but also one who is above time, whose existence is known to us only through his action. The *prae* of God's action: this means not just the preeminence of history over metaphysics but also the rejection of a purely existential version of the gospel message—quite simply because the gospel message means the primacy of the "in itself" over the "for me", because it excludes the intermingling of the "in itself" with the "for me" that was introduced by Luther and reached its utmost radicality in existential theology; that was eventually forced to conclude that there is no "in itself" outside the "for me", so that, ultimately, the existential interpretation becomes identified with what is interpreted. To seek another independent reality behind it would be foolish objectivism. God acted; this was said before anything was said about man, about his sin, about his search for a gracious God. Thus, the *prae* of God's action means, ultimately, that actio is antecedent to verbum, reality to the tidings of it. In other words, the level of reality of the revelation event is deeper than that of the proclamation event, which seeks to interpret God's action

in human language. Precisely this is the origin of the sacramental principle, the reason why the word of God, which is also action, must be received by man in words and signs".[15]

Ratzinger wants to point out that metaphysical thinking is not the original foundation of reality. The fact that theology needs metaphysics does not mean that it abandons what is primary and fundamental to it. This, in turn, is the actual and concrete action of God in history. If theology enters into a historical symbiosis with metaphysics, it is not in order to become its slave but in order to express through it that divine action that precedes and grounds everything. This is where one can find consistency in the arguments of Tengelyi and Ratzinger: both, and each from his own perspective, defend the possibility and necessity of metaphysical thinking, but only that which is based on a turn to reality as it appears. For the phenomenologist, of course, this means a return to the openness of things, to their giving of themselves; for the theologian, this means an open return to Revelation, which factually and eventually occurred in concrete history. In both cases, what matters above all are the categories of actual experience of reality, not as abstraction and idea but as fact and concrete. Just as Tengelyi defends the possibility of phenomenological metaphysics, Ratzinger opens theology to such a phenomenologically oriented metaphysics. It is about metaphysics sensitive to the historical experience of everyday life revealed by the phenomenological method, as well as about phenomenology open to metaphysics and seeking the truth of things.[16]

According to Ratzinger, theology finds in metaphysical formulas the possibility of a deeper expression of the meaning, permanence, and importance of its conviction of God's action in history, which opens perspectives broader than just historical. In his view, theology here becomes the beneficiary of metaphysics. Metaphysics itself, however, receives new horizons and perspectives in this process of adaptation to the needs of theology. Theology also turns out to be a servant of metaphysics.

## 4. Modality of Trinitarian Discourse for Metaphysic

Thinking in the spirit of the gospel has never created, or even had the ambition to create, any particular metaphysics. This, of course, does not mean that, globally speaking, Christianity has no metaphysical significance.[17] The importance of Christianity for understanding reality can be understood both in terms of the history of metaphysics and its substantive intuitions. Christianity is the art of a deeper, ultimate understanding of reality (as Tillich would express it). Christianity and its theology do not, therefore, create any new metaphysics but merely correctly put it in the right light and complete what was lacking in the classical vision of metaphysics.

In the previous paragraph, we analyzed fragments of the New Testament Trinitarian discourse, which, revolving around the analysis of the basic dimensions of Jesus' life (being, possession, and relationship), led to the formation of a certain metaphysical vision. This vision is implicitly included in the events of Jesus and has a concrete meaning for the understanding of the whole reality. Is this approach, however, an expression of the mythological saturation of metaphysics with theology, which has been described as onto-theology?

As is well known, onto-theology essentially means two interconnected forms of thinking. The first of these is the assumption that God is an entity, and it is an entity of the highest quality, which in itself exhausts the whole being. When combined with the nominalistic theory of *univocitas entis*, it leads to a powerful distortion in our understanding of God and the world. In the case of the latter, it is a matter of giving metaphysics a theological form, for if God is the supreme of beings, then it is also the privileged object of metaphysics, in which it exhausts all its ambitions. The proposal to treat fragments of the Trinitarian discourse as a creative background for metaphysics, however, operates in a completely different field of understanding than the two principles mentioned above. They are not rules of either the Trinitarian discourse itself or its analogous application to the description of the reality of the world. Fragments of the Trinitarian discourse treat the existence, possession, and relations of Jesus only as a perspective projection of the mystery of God into the history of man, which makes the fragments of the finite reality become

icons of that mystery. In this iconic projection, the metaphysical understanding of the world is simultaneously broadened. Without the reification and objectification of God and the exaggerated and erroneous deification of the world, the possibility of a simultaneous understanding of God and the world opens up. The guardians of iconic projection are the strategies of apophaticism and analogy. Thanks to them, as far as they belong to the very nature of Trinitarian discourse, it is possible to open metaphysics to theology without falling into onto-theology. Such a way of proceeding and thinking reverses onto-theology's basic presuppositions and produces what could be called theo-ontology.

Let us return, however, to the essence of the presented argument, which affirms the modality of the fragments of the Trinitarian discourse for metaphysics. The subsidiarity of the Trinitarian discourse for metaphysics becomes visible already in the long process of thinking into what Jesus bears witness to in his way of being, possessing, and entering into a relationship. It is enough to state that already in the fourth century, the category of relations entered the theological dictionary permanently as the deepest and most basic category describing the newness that appeared in the world with Jesus and which was already noticed and initially described by Irenaeus of Lyon. The great theological debates about dogma at Nicea (325), which lasted at least until 381, led to a real revolution in classical metaphysics.[18] The interpretations of the apostolic witness in the shape of the Nicean doctrine of co-substantiality (*homoousios*) break the static and monadic vision of metaphysics, in which the most basic tissue/cell/element of reality is substance. The Fathers who reflected on the apostolic witness of God revealed in Jesus, had begun to understand that it is not the substance but the relationship that is the most fundamental level of the whole reality. Relationship thus became the main category for the interpretation of the existence and possession of Jesus and, thus, of the whole renewed vision of metaphysics.

The concepts of being, possessing, and entering into relationships turn out to be largely interdependent, and the background and measure of this mutual dependence is the relationship. It is the relationship that begins to determine the content of existence and possession. From such a theological perspective, introduced during the Arian controversy, the basic task of metaphysics will be to explore existence and possession in such a way as to reveal their relational structure. Existence and possession encounter an explanation of their own depth only in relation. Metaphysics, in which the key words are existence, possession, and relationship, is the intuition of the fundamental character of the latter.

This approach seems to be in line with our everyday experience, in which being, possession, and relationship overlap. Let us consider, for example, the following sentence: *have a child*. What is this sentence about? What is the proper meaning of a verb *to have* in this case? It is obvious that people are not possessed as things are. The statement *have a child* defines more than the state of possession; it defines the existence of the subject of the statement. *Having* a child means *being* a parent. In the statement *have a child*, it is not so much the structure of having as the structure of being that appears. What is more, it is being in a state of relationship: as far as the sentence *have a child* means and identifies being a parent, it also defines being as a being in relation. The same happens in the case of the category of existence, which is a form of possession, but also originally, as evidenced by Heidegger, a form of relation (coexistence, *mit-sein*). In this way, my simple *have* identifies me as the owner of myself (cf. Heidegger 1993, 165f.) and sets me in relation to others. It seems that Heidegger's analysis of *Dasein* brings all these threads into the light, of course, in an insufficient way. Going further, it can be said that on the level of relations, it is also possible to find similar references to existence and possession. And it is not only about the fact that in the realm of contingency, the relation seems to be secondary to existence; it is also about the fact that it is indeed co-extensive with it. Relationship is always the name that names existence and possession.

Can such radically relational thinking, which reforms the understanding of existence and possession in their organic and generic mutuality, become a mental form (*forma mentis*) of metaphysics? In the case of the mystery of God's inner life revealed in Jesus, this type of thinking, expressed most deeply by Thomas in his definition of the divine person (*relatio*

*ut subsistens*[19]), does not cause much difficulty in being accepted. The metaphysics of the Trinity is based on the idea of relation understood, from the fourth century onwards, not as an accident but as a substance (cf. Maspero 2018, pp. 255–84). In the case of the metaphysics of finite existence, however, the matter seems a little less obvious. This is due to the fact that the scheme proposed in Revelation calls for going further than classical metaphysics allows. If in the case of the divine reality given to us in Revelation (that is, as it is given to us), the change of substance into a relation—and this is what the Trinitarian discourse is all about—seems less glaring (after all, the logic of divine life goes beyond our understanding), then it raises questions in the case of being transferred to the metaphysical description of our world. First of all, are the beings of our world pure relations? It should be noted that the metaphysical perspective presented here does not assume such a reduction of substance to the relation in the contingent realm. Instead, it points to the simple fact that the more we explore the acts of existence and possession, the more the relational structure of reality reveals itself to us, and consequently, the more we understand the essence of the metaphysics of exodus, namely the truth that beings exist as much as they ecstatically enter into relations. Metaphysics is nothing more than an articulation of such a state of affairs. In this respect, one of its most promising versions is the phenomenological reduction of appearance to givenness developed by Jean-Luc Marion (2002, pp. 39–53). While it indicates the ecstatic nature of things (their ability to give oneself), it reveals, in everything that appears, the relational infrastructure.

Although the above arguments are based on speculation, they do not distract us from the common everyday experience of what is its most fundamental level, namely love. Love itself is the very core of the Paschal metaphysics revealed in fragments of the Trinitarian discourse. It is adequately expressed in the form of a poetic question that expresses the unity of being, having, and relating: *if you do not exist, tell me why I should exist*? This question ingeniously expresses the depths of classically defined metaphysics as the science of being. If metaphysics is to articulate being as being, it must become an open thematization of relations. Otherwise, it will remain only an empty theory of monadic objects, a kind of system that cheats us with the openly illusionary propositions of sense.

## 5. Instead of Conclusion: Mystery Opened Sacramentally in Fragments

The aforementioned arguments, deriving their power from a specific case of Trinitarian discourse in its genetic origins, lead to reflection on the fragmentation of language and thinking in general. What is the meaning of the fragment? What is its role in thinking and living processes? Does the fragmentary nature of metaphysics mean that we are deprived of the possibility of thinking about the whole in advance? Is fragmentation, as a sign of finiteness, condemned to pessimistic interpretations, which make an argument for the impossibility of reaching the truth and propose any deepened vision of the whole? In response to these questions, a few comments on the fragmentarity of theology, ontology, and epistemology should be made.

(1′) The Trinitarian discourse of Christianity emerges from fragments of the description of Jesus and His lifestyle. This style reveals the identity of Jesus, who lives his existence, possessions, and relationships as a pilgrim on the road. In the way he exists, possesses, and enters into relationships, the mystery of divine existence is revealed. This is why Christians recognize in Jesus, according to their apostolic experience, the final unveiling of the mystery of the divine in and through relationships. This revelation is the basis for the Christian revision of metaphysics, in which the main category becomes relation instead of substance. Its effects are multidimensional and include not only metaphysics but also the theory of cognition. The fragmentary nature of the Trinitarian discourse, as the real way in which God reveals Himself, demands the adoption of the apophatic principle in epistemology governed by analogy.

(2′) Previous considerations led us to conclude that metaphysics as a project is always fragmentary. Metaphysics is not some kind of total, extra-contextual, and holistic knowledge *in se*. In metaphysics, we can only get to know the whole through its fragments.

Nevertheless, these fragments are actually fragments of the whole. All metaphysics characterized by an unrestrained drive for an absolute vision of the whole should be considered chimeras of true thinking. They are not in line with the real nature of the world described in them. Such systems are, for example, metaphysics proposed by German idealists, above all Hegel. In fact, they are a form, as Balthasar would probably express it, of an assault of gnosis on love. In their own nature, they represent the ambitions of reason for total control over known reality. Thus, they generically contradict the apophatic character of the discourse, which is characteristic of Christianity. Their criticism, from Kierkegaard to Levinas, based on Judeo-Christian threads of the hermeneutics of reality should be considered generally justified.

(3′) Christianity has a positive view of the fragment[20], both in metaphysics and in epistemology. Fragmentedness opens up the possibility of grasping the whole. Indeed, we have no other access to the vision of the whole but only through its fragments. It is not, therefore, a barrier that stands in the way of a perfect, full being and knowledge, but, on the contrary, the possibility of opening it up in a procedural way.

We should recall here another metaphysical act that took place in the fourth century, the subject of which was the debate on infinity (Mühlenberg 1966). For the Greeks, it meant pure imperfection, and from the perspective of the gospel and its witness to the inner life of God, it gradually took on a positive character, signifying the totality of the divine actual fulness as well as the possibility of opening the ever-growing and progressing access to divine infinity by the finite man. For Gregory of Nyssa, the finiteness of man is the basis of the never-ending path of knowledge and love of the Infinite. Thus, there is a kind of correspondence between real infinity and finiteness, between the whole and the fragment. What is finite and fragmentary is potentially open to infinity and the whole. All this may sound too abstract.

We must remember, however, that what is at stake in the ancient debate on "infinity" is not an abstract speculative game but a contemplation of mystery. Its content is the reality of the Trinitarian giving of oneself to a man in the history of Jesus, who accepted the fragmentary nature of our existence in order to express the mystery of the divine way of existence through it and in it. In the various stages/fragments of His life, as the hymn of the Liturgy of the Hours for the solemnity of The Epiphany indicates, "there appears a being that has no beginning or end, *antiquis celo et chao*". Christian contemplation of God in the child Jesus indicates the paradox described here. The Trinitarian giving of oneself to the world in fragments saves the fragmentation and finiteness that so deeply characterize our human, created condition. At the same time, it gives it a new meaning. It makes it possible that our own finiteness and fragmentation may become the Trinitarian way of the divine coming to us. Moreover, it enables our own finite and fragmentary experiences of existence, possession, and relationship to open up to us a mysterious way of divine existence, so that they become signs of mystery.

It was Ferdinand Ulrich who best expressed this paradoxical nature of fragmentation when, thinking about creation, he saw its metaphysical beginning in "love as a movement that makes existence finite" (Ulrich 2018, pp. 47–60). The finiteness, and therefore the fragmentation of existence, are signs of divine love for "what is not that it might be". God, here again, says Paul, "subjected everything to vanity in hope". Finiteness, futility, and fragmentation come from divine love and hope. Does this not give them a completely different character? For it is in them that we see the whole of the Trinitarian love and hope from which we all come and which invites us constantly to return to the paternal source of existence. In this way, fragmentation becomes a primary blessing, both metaphysically and cognitively, as long as it is an invitation to enter into a relationship with the Father, Son, and Holy Spirit.

(4′) It turns out that the fragmentary nature of Jesus' life, which first creates a fragmentary Trinitarian discourse and then defines the foundations of the Christian revision of classical metaphysics, becomes for us a place of understanding and renewal of our experience. But what does our access to those fragments of Jesus' life in which the truth of God

and the world are revealed at once look like? In short, Jesus' way of existence, possession, and relationship, in which the divine mystery is revealed and given, continues for us in the Living Church and in her sacraments. Sacramental logic is a reflection of the Paschal metaphysics described above. The sacraments, by taking us into the situation of Jesus' life[21], in His way of being, possession, and relationship, open us up in their fragmentation to the whole Trinitarian love given to us by God. From this perspective, our participation and its always partial nature (participation as *partem-capere*, as taking a part, a fragment) opens up the possibility for us to enjoy the always greater divine whole, an enjoyment spread over time and eternity.

**Funding:** This research received no external funding.

**Data Availability Statement:** Not applicable.

**Conflicts of Interest:** The author declares no conflict of interest.

## Notes

1   This text intentionally does not take a transcendentalist perspective. *Modus* therefore means a certain/concrete way of being and revealing oneself. The term can also be applied in a formal sense: a concept is modal in the sense of being an expression and carrier of meaning. In this sense, it is about operational categories (conceptual, instrumental tools). Of course, such categories should correspond to and originate from the very reality they are used to describe.

2   English translation: (Barthes 2010).

3   Cf. Roland Barthes (2002, p. 670): "Écrire par fragments: fragments sont alors des pierres sur le pourtour du cercle: je m'étale en rond: tout mon petit univers en miettes; au centre, quoi? Son premier texte ou à peu près (1942) est fait de fragments; ce choix est alors justifié à la manière gidienne „parce l'incohérence est preferable à l'ordre qui déforme". Depuis, en fait, il n'a cessé de pratiquer l'écriture courte: tableautins des Mythologies et de l'Empire des signes, articles et préfaces des Essais critiques, lexies de S/Z, paragraphes titrés du Michelet, fragments du Sade Iet du Plaisir du texte". Cf. Adam Dziadek (2004, 106nn).

4   This essay presupposes an idea of revelation proper to the tradition of IVatican Council. Revelation is conceived as auto-communication (Rahnerian *Gottes Selbstmitteilung*). For more details cf. Magnus Lerch (2015).

5   Jean-Luc Marion (2008, pp. x–xi; 2016); and in the very special way Id., Marion (2020, pp. 155–267).

6   One of the main convictions of the argument presented here is the incompleteness of transcendentalist-style thinking which ulimately can be a form of disguised docetism. Hence my emphasis on concrete history as a space of revelation. The fact that this concrete history is treated as a sign of supra-historical reality (transcendent not transcendental) does not at all remove its importance, significance, autonomy. That is why Jesus simultaneously reveals with his life (existence, possession, relation) humanity and the essential contours of divinity.

7   The category of memory becomes more and more fundamental in latest systematic Christology and dogmatics. For the biblical roots of it cf. Dunn (2003, pp. 129, 173, 186). The category of memory is crucial for Richard Bauckham (2006).

8   Throughout this article, the two terms are treated as synonymous. This is a kind of simplification of a huge problem, one of the main ones facing metaphysics and its history. I allow myself this simplification because the subject of this contribution is not a theory of being as such, but only the application of the existential categories inherent in the life of Jesus to Trinitarian theology by means of a phenomenological description. Thus, when I treat of existence I mean the categorical experience of Jesus as it is described in Scripture, in its openness to the being revealed in it. Thus, the categories of existence and being become synonymous, especially on the basis of the openness of Jesus' concrete existence toward the unveiling of being and its truth.

9   For Ratzinger there is a direct link between *ego eimi* and the later Christological dogmas of the Church. Cf. Joseph Ratzinger (2014, pp. 832–49).

10   An important intuition and also a description of such a metaphysics can be found in Joseph Ratzinger (2020, pp. 89–98).

11   On the concept of co-existence, co-being cf. Yann Vagneux (2015).

12   On the problem metaphysical nature of possession cf. A. Workowski (2009). An important result of the book is the confirmation of the conjecture that our fascination with possession is not an eccentricity, but it is rooted in the fundamental structures of the world and in our own constitution.

13   Balthasar shows the structure and nature of Jesus' possession based on His way of experiencing time, see his *A Theology of History* (von Balthasar 1994, p. 41): "Thus the whole basis of time for the Son is his receptivity to God's will. In that receptivity he receives time from the Father, both as form and as content. The time that he receives is the Father's time, qualified moment by moment. For him, there is no such thing as "time in itself"; what might seem to be this, in his assumption of "human nature in itself", is, in the very act in which he assumes it, subjected to and incorporated into the uniqueness of his Sonship. Where he is concerned, there is no such thing as empty time, available for filling with some indifferent content or other. To have time means, for him, to have time for God, and is identical with receiving time from God. Hence the Son, who has time, in the world, for God, is the

point at which God has time for the world. Apart from the Son, God has no time for the world, but in him he has all time. In him he has time for all men and all creatures: in relation to him it is always Today".

14 Cf. project of Trinitarian ontology in Piero Coda (2012, pp. 159–79); cf. Emmanuel Prenga (2018). The accurate systematization of the initiative of so called "trinitarian ontology" is available in Eduard Fiedler (2021, pp. 101–24). Fiedler distinguishes eight types of this ontology: relational, connected with the notion of gift (as in Milbank), phenomenological, Thomistic, critical toward metaphysics, apophatic, Hegelian, contemplative.

15 Joseph Ratzinger (1989, pp. 185–86). It should be remembered here that in the last footnote of his contribution to the question of the relationship between history, eschatology and metaphysics, Ratzinger indicates that his thinking is evolving towards a more balanced account of the "prae" of divine action in history that is relevant here, and its relation to metaphysics. At the same time, however, he chooses to leave this "prae" in the body of the text. It seems that the whole logic of the final footnote in question is only to ultimately prevent his proposal from being interpreted in terms of empty and sterile historicism. Cf. Ratzinger, *Principles of Catholic Theology*, 190: "In view of the fundamental meaning of this "is", I would stress more strongly today than I have in these pages the irreplaceability and preeminence of the ontological aspect and, therefore, of metaphysics as the basis of any history. Precisely as a confession of Jesus Christ, Christian faith—and in this it is completely loyal to the faith of Abraham—is faith in a living God. The fact that the first article of faith forms the basis of all Christian belief includes, theologically, the basic character of the ontological statements and the indispensability of the metaphysical, that is, of the Creator God who is before all becoming". Cf. also, Joseph Ratzinger (2004, p. 228): "precisely because this "being" is no longer separable from its *actualitas*, it coincides with God and is at the same time the exemplary man, the man of the future, through whom it becomes evident how very much man is still the coming creature, a being still, so to speak, waiting to be realized; and what a short distance man has even now progressed toward being himself. When this is understood, it also becomes clear why phenomenology and existential analyses, helpful as they are, cannot suffice for Christology. They do not reach deep enough, because they leave the realm of real "being" untouched".

16 In fact, in Ratzinger, such a phenomenologically colored metaphysics takes the shape of a new theology of existence. Ratzinger's existentialism, so visible throughout his work, is based on the conviction of the symbiosis of metaphysical, historical and theological thinking. Cf. Tomasz Samulnik (2022). Cf. Michel Deneken (2011, pp. 499–510).

17 It was emphasized by Klaus Hemmerle in his famous *Thesen*. English translation: Klaus Hemmerle (2020).

18 Ratzinger, *Introduction to Christianity*, 184: "Therein lies concealed a revolution in man's view of the world: the sole dominion of thinking in terms of substance is ended; relation is discovered as an equally valid primordial mode of reality. It becomes possible to surmount what we call today "objectifying thought"; a new plane of being comes into view. It is probably true to say that the task imposed on philosophy as a result of these facts is far from being completed—so much does modern thought depend on the possibilities thus disclosed, without which it would be inconceivable". The development of the relational metaphysics in the theology of Cappadacions is presented in: Jaroslav Pelikan (1995), and Giulio Maspero (2013).

19 Thomas Aquinas. *Summa Theologica*. I, 29, 4: "Persona igitur divina significat relationem ut subsistentem".

20 It is visible in Balthasar treatment of the difference and otherness, cf. Hans Urs von Balthasar (2004), kindle position 2029.

21 Leo the Great, (Sermo. 74, 2: PL 54, 398): "what was visible in our Savior has passed over into his mysteries".

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
