# Peer review of "Phenomenological Fragments of Trinitarian Discourse: Being, Having, Relating"

_religions, doi:10.3390/rel14070929_

Round 1

Author Response

Overall. The paper studies three main “existential operators” used in a Trinitarian analysis. The term “modal” is used here, but it’s unclear whether or not the better term would be” transcendent.” The operators are understood to be “conceptual,” but it would be necessary to call them more precisely something like “limit concepts” when applied to a transcendent reality such as the Trinity.

My text intentionally does not take a transcendentalist perspective. Modus therefore means a certain way of being and revealing oneself. Conceptual operators - the expression indicates the instrumentality of the analyzed concepts to the extent that they refer to actual modes of being. Of course, such concepts are always analogous in theology and hence also liminal. 

The paper also needs to clarify whether existence is metaphysical or not: the author seems to assume that it is, but that demands an argument since to many philosophers “existence” can only be problematically equated with “being.” Related to this, more time in the paper needs to be taken to analyze the ontotheology problem introduced by Heidegger and others.

As the reviewer points out, the open problem is a huge one, and I don't think there is room in my article to touch on it even a little. My take is situated in the field of realism. This is how I understand phenomenology after all, as a return to being, to history as an essential dimension of reality. 

In the “relationship” existential operator as applied to Jesus, the relationship between Jesus and the Father should be shown to be absolutely primary, at least Scripturally.

Absolutely yes, I agree. 

Are the terms “paschal metaphysics” and “paschal ontology” equivalent for the author? This raises the earlier comment about the need to clarify “being” and “existence.”

Thank you for the reminder. I was treating both terms synonymously. However, ontology is a modern mutation of the metaphysics program. I will standardize on "paschal metaphysics". 

Well researched on contemporary discussions of the Trinity. But it would be good to have more citation of Marion’s work, especially in Givenness and Revelation, since his current and important --but quite different from the author’s -- Trinitarian analysis is mentioned but not engaged with sufficiently.

I analyze Marion's Trinitarian theology in another contribution. Here, as the reviewer notes, I take a slightly different path than Marion (his analyses are more concerned with transcedental structures and mine with the categoriality of Jesus' life story). 

Editing needed, mainly regarding a few typos. Homoousios in the development of Trinitarian doctrine should be equated not with “co-existent” but “co-essential.”

I have amended in reference to Nice. Thank you for bringing it to my attention. I left in reference to Jules Monchanin's concept. 

Refers to Heidegger using existence is a possession. This would have to be clarified with a reference to his writings. 

I have added a footnote to Grundprobleme der Phanomoenologie, pp. 165ff (Michselbsthaben).

Reviewer 2 Report

The author has tried to put forward a new method for carry a meaningful exploration of Trinitarian Discourses. His chosen method is phenomenological because it is starting points is the fragments of the life and works Jesus Christ is a revelation of the nature of the Trinitarian God. I would say that his quest is an ascent to Economic Trinity from the Christology from below. However, ever it is not clear whether that adequately accounts for the Immanent Trinity. Be that as it may, it seems the author does not acknowledge the limitation of the phenomenological method. Whether is from Husserlian, Heideggerian or Marion's perspective, Phenomenology is limited and can at best account for perceptive (empirical) beings.

If phenomenology is descriptive by nature, it means that it would be difficult for phenomenology to be the best method to use in the study revelation or theology. This is because the objects that revelation study are not empirical. Revelation and theology in general would require explanatory and exploratory methodology since it language is principal analogous. The Husserlian dictum "to things themselves" which captures his conception of phenomenology is not as simple as it might be at first instance. For Heidegger, phenomenology answers only "how" question but not "what". Hence phenomenology would presupposed that the "what" question is settled but that is not the case with Trinity. 

The author's beginning from line 83  seems to be a simplistic. The passage highlights that experience of the disciples is an empirical experience of the Word incarnate. That in itself is an interpretation of divine revelation.

Returning to the words and works of Jesus the Christ does not imply that the meaning of revelation rests on the givenness of the given. The return is always a quest for a understanding of the given and subsequently affirmation of that which is understood. As phenomenology is in adequate to method to account for revelation. I recommend that author should see Lonergan's Transcendental Method or Generalised Empirical Method (Method in Theology chapter one or Insight: A Study of Human Understanding).

The connection between the ego eimi and the Horeb experience is not phenomenological but explanatory.

Line 192: Do the Father and Holy Spirit constitute life of Jesus or do the three constitute the Holy Trinity?

Line 209 ff: What the author claims to be description or phenomenological analysis is in essence an explanation. It is important that the author differentiate between phenomenological method and explanatory method.

Line 286: What exactly does "reality as it appears" means. Is it different from reality as it is?

Line 351: Homoousios in the Nicean creed means oneness of substance or the same substance. So for the translate it as co-existence is forced to make it fit his argument.

Line 357: This needs to be established. The author's position is at best begging the question. The concept of being, possessing, and entering into relations are not synonymous. For their to relationship there should at least two independent or interdependent beings.

Line 374 - 376: Appealing to Heidegger's concept of mit-sein does much to establish the author's position. Dasein is mit-sein (being-with) because it existence in relation with other Daseins, who are completely independent beings. However, the case of the Trinity, the three persons - Father, Son and the Holy Spirit - are one in Godhead.

Line 382: Relationship, existence and possession as used by the author are attributes or accidental categories in the language of Aristotle. An attribute is always an attribute of a being or "substance". So the author needs to show without equivocation the three attributes defines the Trinity.

lines 386-389: he author has not clear pointed out that the issue in question in Thomas is about the definition or meaning of person in relation to divine since three are three person but one divine nature. In saying that divine person signifies "relation as subsisting", Aquinas is unambiguous that it is the signification of the the term person that is issue in question. That is why he continues to write:  "And this is to signify relation by way of substance, and such a relation is a hypostasis subsisting in the divine". So it is person that is relation as subsisting and not relationship per se.

Line 413  seems to be an equivocal understanding of being. Metaphysics as the science of being qua being highlights the that being is that which is and the act of existing.

Lines 434 and 435: It is impossible to talk of relation with presupposing substance or substances. Even in the Christian vision, relation is important because of the divine substance or divine nature in which there are persons in relation.

The quality of English language is very good. Nonetheless, the author needs there about two instances in which the definite article "the" is used where it needs to be remove. Also Ricoeur is spelt incorrectly.

Author Response

The author has tried to put forward a new method for carry a meaningful exploration of Trinitarian Discourses. His chosen method is phenomenological because it is starting points is the fragments of the life and works Jesus Christ is a revelation of the nature of the Trinitarian God. I would say that his quest is an ascent to Economic Trinity from the Christology from below. However, ever it is not clear whether that adequately accounts for the Immanent Trinity. Be that as it may, it seems the author does not acknowledge the limitation of the phenomenological method. Whether is from Husserlian, Heideggerian or Marion's perspective, Phenomenology is limited and can at best account for perceptive (empirical) beings.

If phenomenology is descriptive by nature, it means that it would be difficult for phenomenology to be the best method to use in the study revelation or theology. This is because the objects that revelation study are not empirical. Revelation and theology in general would require explanatory and exploratory methodology since it language is principal analogous. The Husserlian dictum "to things themselves" which captures his conception of phenomenology is not as simple as it might be at first instance. For Heidegger, phenomenology answers only "how" question but not "what". Hence phenomenology would presupposed that the "what" question is settled but that is not the case with Trinity. 

The author is not convinced that the division between theology from below and theology from above is correct. The intention of the article was not to create any theology from below, but only to explore the actual form of Revelation as it happened in history.

The author understands all too well the limits of the phenomenological method. However, he finds it difficult to agree that phenomenology only reaches the phenomenal layer of things. As shown in the article, phenomenology does not mean excluding metaphysics, but only that form of it which remains purely presuppositional, apriori to all experience. Understanding phenomenology as pure transcendentalism or even a method that rejects all metaphysics is, especially from the point of view of its recent history, incorrect. For phenomenology is a path to reality, to being, a path that increasingly learns to respect not only the transcendental requirements of human subjectivity, but above all the very structure and nature of reality (cf. R. Sokolowski).

It is also clear that theology is based on analogy (Lateran IV) and therefore requires explanatory methods. However, is it only such? The author of the article does not suggest that phenomenology is the only appropriate method. His conviction is that Revelation and theology require the whole spectrum of methods and approaches. The possibilities inherent in phenomenology in no way exclude other methods but should be considered complementary to them. It is in this regard, for example, that the author does not share the opposition of some authors (Marion, for example) to metaphysics. However, it is difficult not to concede the point that there is some important connection between Revelation and phenomenology, and that the latter represents some important moment and an important possibility, due to the nature of Revelation, for theology.

The author's beginning from line 83 seems to be a simplistic. The passage highlights that experience of the disciples is an empirical experience of the Word incarnate. That in itself is an interpretation of divine revelation.

Experience is never about nature, but about the person. Since in Christ - according to the definition of faith in Constantinople III - there is only one person, what the apostles experienced is precisely the Eternal Logos, who became man and as such is open to being experienced in and through his own empirical body. John's writings testify to this quite strongly (especially the two prologues).

Returning to the words and works of Jesus the Christ does not imply that the meaning of revelation rests on the givenness of the given. The return is always a quest for a understanding of the given and subsequently affirmation of that which is understood. As phenomenology is in adequate to method to account for revelation. I recommend that author should see Lonergan's Transcendental Method or Generalised Empirical Method (Method in Theology chapter one or Insight: A Study of Human Understanding).

The connection between the ego eimi and the Horeb experience is not phenomenological but explanatory.

I agree that phenomenology in theology does not remove the need for other methods. That is not the point of my text. None of its passages make this point. The reviewer's comments seem to be an interpretation of my approach, from the point of view of a rather narrow understanding of phenomenology. Therefore, he wants to reiterate emphatically that my understanding of the phenomenological model does not in any way invalidate or render unnecessary the explanatory approach inherent in at least the various versions of metaphysics.

Line 192: Do the Father and Holy Spirit constitute life of Jesus or do the three constitute the Holy Trinity?

In the economic order, the life (sic!) of Jesus is constituted by the Father and the Holy Spirit - a matter of the famous Trinitarian inversion (Balthasar). Cf. the baptism scene or the event in the synagogue in Nazareth. This is what the passage deals with.

Line 209 ff: What the author claims to be description or phenomenological analysis is in essence an explanation. It is important that the author differentiate between phenomenological method and explanatory method.

The author's intention is not to reject other methods, but only to point out that one of them - so far little used - can become a properly understood phenomenology.

Line 286: What exactly does "reality as it appears" means. Is it different from reality as it is?

I assume, along the lines of classical realism, that reality appears as it is. Which, of course, does not mean that what appears is its totality.

Line 351: Homoousios in the Nicean creed means oneness of substance or the same substance. So for the translate it as co-existence is forced to make it fit his argument.

Rightly so, I changed the English term.

Line 357: This needs to be established. The author's position is at best begging the question. The concept of being, possessing, and entering into relations are not synonymous. For their to relationship there should at least two independent or interdependent beings.

Yes, it is true.

I changed the sentence: The concepts of being, possessing, and entering into relationships turn out to be largely interdependent, and the background and measure of this mutual dependence is the relationship.

Line 374 - 376: Appealing to Heidegger's concept of mit-sein does much to establish the author's position. Dasein is mit-sein (being-with) because it existence in relation with other Daseins, who are completely independent beings. However, the case of the Trinity, the three persons - Father, Son and the Holy Spirit - are one in Godhead.

Every concept applied to the life of the Trinity is analogous. I did not intend to reduce the substance unity of the Trinity to the idea of community. Anyway, a line further on I state the inadequacy of Heidegger's scheme. I do not refer to it in a blind and naive way. I only wanted to point out that with regard to finite being it is possible to see the connection between being, having and relation. This is important insofar as, the place of the analogous revelation of the Trinity is the finite humanity of Jesus.

Line 382: Relationship, existence and possession as used by the author are attributes or accidental categories in the language of Aristotle. An attribute is always an attribute of a being or "substance". So the author needs to show without equivocation the three attributes defines the Trinity.

My argument here is based on a famous operation carried out especially by Gregory of Nyssa and then, to a somewhat lesser extent, by Augustine and then by Thomas Aquinas, the object of which is to confer the status of substance on accidental categories in the space of attribution to God. In this perspective, being means relationship. Cf. the arguments of Giulio Maspera, especially in “Essere e relazione”. Please also keep in mind that the entire commented passage of my text concerns finite being, which only by analogy becomes the place of Revelation.

lines 386-389: he author has not clear pointed out that the issue in question in Thomas is about the definition or meaning of person in relation to divine since three are three person but one divine nature. In saying that divine person signifies "relation as subsisting", Aquinas is unambiguous that it is the signification of the the term person that is issue in question. That is why he continues to write:  "And this is to signify relation by way of substance, and such a relation is a hypostasis subsisting in the divine". So it is person that is relation as subsisting and not relationship per se.

The entire paragraph clearly indicates that this definition refers specifically to God, and transferring it to the order of creation is not entirely possible. I want to reiterate that the principle of analogy is crucial in understanding my text. At no point do I claim that we have some finite category that is capable of exhausting the mystery of God's life. However, there is, by virtue of creation and incarnation, the possibility of analogical adjudication. Contingent reality, as created, that is, made in image and likeness, sheds light on divine reality, and only in it does it find its ultimate explanation.

Line 413  seems to be an equivocal understanding of being. Metaphysics as the science of being qua being highlights the that being is that which is and the act of existing.

The entire paragraph - clearly, as its beginning suggests - refers not so much to the concept of being, but to its experience, which allows its preconception. 

Lines 434 and 435: It is impossible to talk of relation with presupposing substance or substances. Even in the Christian vision, relation is important because of the divine substance or divine nature in which there are persons in relation.

My text refers directly to the existential analogue, not to the metaphysical argument. The life of Jesus is not metaphysics (for metaphysics is a "meta "discourse, as the name says) - metaphysical intuitions can be formed on its basis. It is from the way of life of Jesus that arguments are gradually drawn to modify the understanding of being and the meaning of existence.

Reviewer 3 Report

The article, inspired mainly by Hans Urs von Balthasar and Joseph Ratzinger, proposes a rereading of the metaphysical dimension of Christian dogma – and of the concept of Metaphysics itself – through articulation with the phenomenological philosophy, especially as developed by Jean-Luc Marion. In this context, an interesting relationship is established between Trinitarian theology and the revelation of the Trinity in the story of Jesus, especially from the perspective of an “Easter ontology” (inspired more by the Gospel of John than by the Synoptics), insofar as the “being” (divine) of Jesus takes place in the concrete event of his existence and his self-awareness: awareness of being, awareness of possession and relationships. This articulation between the revelation of God (metaphysical) and its articulation in the story of Jesus (fragmentary), leads to a re-reading of the relationship between metaphysics and phenomenology, proposing a kind of metaphysics “from below”, that is, from the particular historical event, and not from a priori abstract concepts. Revealing the whole in the fragment, the fragment has historical and epistemological priority.

The article does not present a completely original proposal, however it presents a clear and consistent argument, well founded and with clear and sustained conclusions. And the perspective of a possible phenomenological metaphysics inspired by Marion – with references to Levinas and Richir – seems to me to be very interesting and fertile, as well as the development of a relational ontology, inspired by the Trinitarian revelation (theo-ontology).

For the theological debate with the author (which in no way diminishes the quality of the article, on the contrary) one could ask whether the model of phenomenology – and also of theology – is not too transcendental (in the style of Marion), reducing the value of the historical event, in its concrete historicity. How to consider the revealing value of Jesus' humanity, if his human activity is immediately absorbed in his divine being, as possession of something, in relation to the Father and the Spirit? Even looking in an interesting way to combine the immanent Trinity with the economic Trinity, it seems that the revelation is confined to the Trinitarian process (ontologicaly). How to consider, in this revelation, the revelation of true humanity – and not just of divinity? But if the revelation of humanity (and of a human God) is not heeded in the story of Jesus, how can Jesus be followed by humans? In short, aren't there traces of docetism in this transcendental-metaphysical reading of the phenomenology of revelation in Jesus?

Author Response

For the theological debate with the author (which in no way diminishes the quality of the article, on the contrary) one could ask whether the model of phenomenology – and also of theology – is not too transcendental (in the style of Marion), reducing the value of the historical event, in its concrete historicity. How to consider the revealing value of Jesus' humanity, if his human activity is immediately absorbed in his divine being, as possession of something, in relation to the Father and the Spirit? Even looking in an interesting way to combine the immanent Trinity with the economic Trinity, it seems that the revelation is confined to the Trinitarian process (ontologicaly). How to consider, in this revelation, the revelation of true humanity – and not just of divinity? But if the revelation of humanity (and of a human God) is not heeded in the story of Jesus, how can Jesus be followed by humans? In short, aren't there traces of docetism in this transcendental-metaphysical reading of the phenomenology of revelation in Jesus?

One of my main convictions is the incompleteness of transcendentalist-style thinking, which, as the reviewer rightly states, can be a form of disguised docetism. Hence my emphasis on concrete history as a space of revelation. The fact that this concrete history is treated as a sign of supra-historical reality (transcendent not transcendental) does not at all remove its importance, significance, autonomy. That is why Jesus simultaneously reveals with his life (existence, possession, relation) humanity and the essential contours of divinity.